# Flipping between Polycomb repressed and active transcriptional states introduces noise in gene expression

Gozde Kar[1], Jong Kyoung Kim[1,2], Aleksandra A. Kolodziejczyk[1,3], Kedar Nath Natarajan[1,3], Elena Torlai Triglia [4], Borbala Mifsud [5,6,7], Sarah Elderkin[8], John C. Marioni[1,3,9], Ana Pombo [4] & Sarah A. Teichmann[1,3]

Polycomb repressive complexes (PRCs) are important histone modifiers, which silence gene expression; yet, there exists a subset of PRC-bound genes actively transcribed by RNA polymerase II (RNAPII). It is likely that the role of Polycomb repressive complex is to dampen expression of these PRC-active genes. However, it is unclear how this flipping between chromatin states alters the kinetics of transcription. Here, we integrate histone modifications and RNAPII states derived from bulk ChIP-seq data with single-cell RNA-sequencing data. We find that Polycomb repressive complex-active genes have greater cell-to-cell variation in expression than active genes, and these results are validated by knockout experiments. We also show that PRC-active genes are clustered on chromosomes in both two and three dimensions, and interactions with active enhancers promote a stabilization of gene expression noise. These findings provide new insights into how chromatin regulation modulates stochastic gene expression and transcriptional bursting, with implications for regulation of pluripotency and development.

[1] European Molecular Biology Laboratory-European Bioinformatics Institute (EMBL-EBI), Wellcome Trust Genome Campus, Hinxton, Cambridge CB10 1SD, UK. [2] Department of New Biology, DGIST, Daegu 42988, Republic of Korea. [3] Wellcome Trust Sanger Institute, Wellcome Trust Genome Campus, Hinxton, Cambridge CB10 1SA, UK. [4] Epigenetic Regulation and Chromatin Architecture Group, Berlin Institute for Medical Systems Biology, Max Delbrück Center for Molecular Medicine, Robert Roessle Strasse, Berlin-Buch 13125, Germany. [5] Cancer Research UK London Research Institute, 44 Lincoln's Inn Fields, London WC2A 3LY, UK. [6] Department of Genetics, Evolution and Environment, University College London, Gower Street, London WC1E 6BT, UK. [7] William Harvey Research Institute, Queen Mary University London, Charterhouse Square, London EC1M 6BQ, UK. [8] Nuclear Dynamics Programme, The Babraham Institute, Babraham Research Campus, Cambridge CB22 3AT, UK. [9] Cancer Research UK Cambridge Institute, University of Cambridge, Li Ka Shing Centre, Robinson Way, Cambridge CB2 0RE, UK. Correspondence and requests for materials should be addressed to S.A.T. (email: st9@sanger.ac.uk)

Embryonic stem cells (ESCs) are capable of self-renewing and differentiating into all somatic cell types[1, 2], and their homeostasis is maintained by epigenetic regulators[3]. In this context, Polycomb repressive complexes (PRCs) are important histone modifiers, which play a fundamental role in maintaining the pluripotent state of ESCs by silencing important developmental regulators[4]. There are two major PRCs: PRC1, which monoubiquitinylates histone 2 A lysine 119 (H2Aub1) via the ubiquitin ligase RING1A/B; and PRC2, which catalyzes dimethylation and trimethylation of H3K27 (H3K27me2/3) via the histone methyltransferase (HMT) EZH1/2.

Recently, we discovered that a group of important signaling genes coexists in active and Polycomb-repressed states in mouse ESCs (mESCs)[5]. During the transcription cycle, recruitment of histone modifiers or RNA-processing factors is achieved through changing patterns of post-translational modifications of the carboxy-terminal domain of RNAPII[6]. Phosphorylation of S5 residues (S5p) correlates with initiation, capping, and H3K4 HMT recruitment. S2 phosphorylation (S2p) correlates with elongation, splicing, polyadenylation, and H3K36 HMT recruitment. Phosphorylation of RNAPII on S5, but not on S2, is associated with Polycomb repression and poised transcription factories, while active factories are associated with phosphorylation on both residues[5, 7, 8]. S7 phosphorylation (S7p) marks the transition between S5p and S2p[9], but its mechanistic role is unclear presently.

Our genome-wide analyses of RNAPII and Polycomb occupancy in mESCs identified two major groups of PRC targets: (1) repressed genes associated with PRCs and unproductive RNAPII (phosphorylated at S5 but lacking S2p; PRC-repressed) and (2) expressed genes bound by PRCs and active RNAPII (both S5p and S2p; PRC-active)[5]. Both types of genes are marked by H3K4me3 and H3K27me3, a state termed bivalency[1, 10]. H3K4me3 correlates tightly with RNAPII-S5p[5], a mark that does not distinguish PRC-active and Polycomb-repressed states.

The role of PRCs in modulating the expression of PRC-active genes was shown by PRC1 conditional knockout (KO). Sequential ChIP and single-cell imaging showed mutual exclusion of S2p and PRCs at PRC-active genes[5], although PRCs were found to co-associate with S5p. This indicates that PRC-active genes acquire separate active and PRC-repressed chromatin states. It remains unclear whether these two states occur in different cells within a cell population, or within different alleles in the same cell[5]. This pattern of two distinct chromatin states could imply a digital switch between actively transcribing and repressed promoters within a population of cells, thereby introducing more cell-to-cell variation in gene expression compared to genes with both alleles in active chromatin states.

Motivated by this hypothesis, here, we integrate states of histone and RNAPII modification from a published classification of ChIP-seq data[5] with single-cell RNA-sequencing (RNA-seq) data generated for this analysis. The matched chromatin and scRNA-seq data sets allow us to decipher, on a genome-wide scale, how differences in the chromatin state can affect transcriptional kinetics. A schematic overview of our analysis strategy is shown in Fig. 1. We focus on active PRC-target genes that are marked by PRCs (H3K27me3 modification or both H3K27me3 and H2Aub1) and active RNAPII (S5pS7pS2p), and compare these with "active" genes (marked by S5p, S7p, S2p without H3K27me3 and H2Aub1 marks). We quantify variation in gene expression and transcriptional kinetics statistically and by mathematical modeling (Fig. 1). In addition, we map the functions of PRC-active genes in the context of pluripotency signaling and homeostasis networks. Further, we analyze the linear ordering and three-dimensional contacts of PRC-active genes on the mouse chromosomes. Finally, we investigate the effect of Polycomb on regulating transcriptional heterogeneity by deletion of *Ring1A/B*, followed by single-cell profiling.

## Results

**Single-cell RNA-seq and data processing**. To investigate how Polycomb repression relates to stochasticity in gene expression, we profiled single-cell transcriptomes of mouse OS25 ESCs cultured in serum and leukemia-inhibitory factor (LIF), previously used to map RNAPII phosphorylation and H2Aub1[5]. Single-cell RNA-seq was performed using the Fluidigm C1 system, applying the SMARTer kit to obtain cDNA and the Nextera XT kit for Illumina library preparation. Libraries from 96 cells were pooled and sequenced on four lanes of an Illumina HiSeq2000 (Fig. 1; please refer to Methods for details).

Next, we performed quality-control analysis for each individual cell data set and removed poor-quality data based on two criteria (as described before in ref. [11]). Cells were removed if: (1) the total number of reads mapping to exons for the cell was lower than half a million and (2) the percentage of reads mapping to mitochondrial-encoded RNAs was higher than 10%. We also compared normalized read counts of genes between cells and found many genes abnormally amplified for three cells. Therefore, we removed these cells, resulting in 90 cells that could be used for further analysis. For these 90 cells, over 80% of reads were mapped to the *Mus musculus* genome (GRCm38) and over 60% to exons (Supplementary Fig. 1A–C).

OS25 ES cells are grown under *Oct4* selection and do not express early-differentiation markers such as Gata4 and Gata6[5], having the expected features of pluripotency. They are ideal for studying Polycomb repression and its impact on transcriptional cell-to-cell variation as compared to other culture conditions such as 2i (serum-free). ESCs grown in 2i show decreased Polycomb repression and RNAPII poising at well-characterized early-developmental genes[12], therefore making 2i conditions the least ideal conditions to study mechanisms of Polycomb regulation in the pluripotent state. As previously shown[5], we do not observe distinct subpopulations of cells based on key pluripotency factors and differentiation markers in our OS25 single-cell data sets (Supplementary Fig. 1D).

In addition, we compared single-cell expression profiles of the OS25 ESCs grown under *Oct4* with recently published scRNAseq data sets from mESCs cultured in serum + LIF and 2i[11]; Principal component analysis using pluripotency genes and differentiation markers shows that OS25 cells are more similar to the subpopulation of pluripotent serum cells, rather than the subpopulation of serum cells that are either "primed for differentiation" or are "on the differentiation path" (Supplementary Fig. 1E).

**Defining chromatin state and gene expression noise**. We integrated our new single-cell RNA-seq data with a previous classification of gene promoters according to the presence of histone and RNAPII modifications[5] (Fig. 1). Comparison of our average single-cell expression profiles with the bulk gene expression (mRNA-seq) profiles from Brookes et al.[5] yields a high correlation (Spearman's rho = 0.87, Supplementary Fig. 1F), suggesting that the chromatin and RNAPII data reflect cells in the same biological state as the single-cell RNA-seq data.

Next, we analyzed gene expression variation within the single-cell data. First, we quantified cell-to-cell variation at each mean expression level using the coefficient of variation (Supplementary Fig. 2A). Cell-to-cell variation can arise either due to stochastic gene expression itself, or technical noise or confounding expression heterogeneity due to biological processes such as the cell cycle.

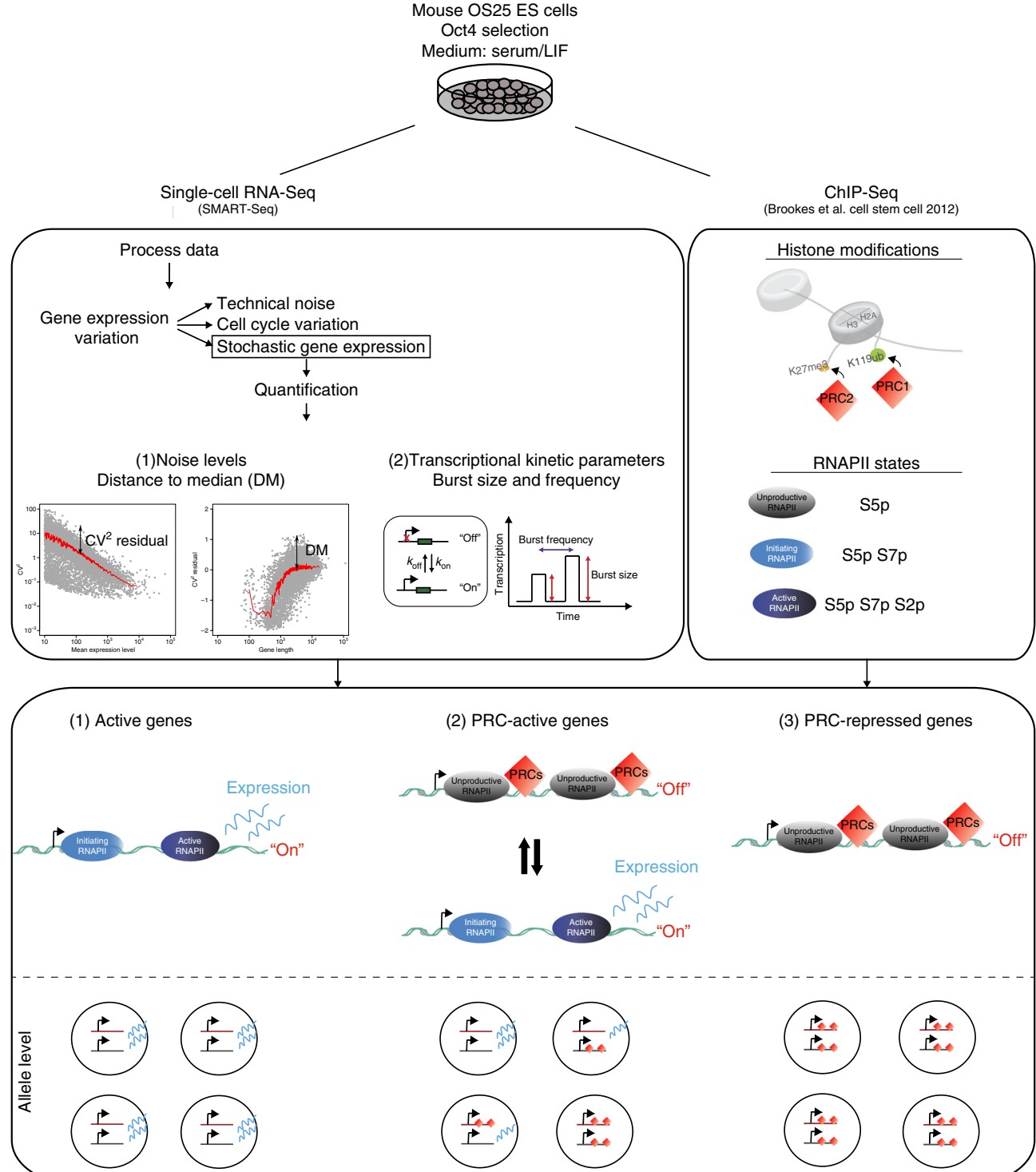

**Fig. 1** Summary of methodology. OS25 mESCs were cultured and characterized by single-cell RNA-seq using the Fluidigm C1 system, applying the SMARTer kit to obtain cDNA and the Nextera XT kit for Illumina library preparation. OS25 cells are grown in conditions that select for undifferentiated cells (high *Oct4*-expressing). Libraries from 96 cells were pooled and sequenced on four lanes of a HiSeq. After quality-control analysis of cells, 90 cells out of 96 remained for further analysis. We first unraveled contributions of components of gene expression variation using the scLVM method[13]. Removing cell cycle variation and technical noise allowed us to focus on stochastic gene expression. Gene expression variation can be quantified by $CV^2$ or DM, which is a measure of noise independent of gene expression levels and gene length. To explore the transcriptional kinetics of OS25 ES cells, poisson-beta model[16] was fitted to single-cell gene expression data, leading to estimates of burst frequency and size. Next, histone and RNAPII promoter modifications were obtained from Brookes et al.[5] and integrated with single-cell RNA-seq to investigate relationship between stochastic gene expression and epigenetics. Active genes with no PRC marks are usually in the "on" state with high burst frequencies ($k_{on}$), PRCr genes are mostly "off" and PRC-active genes switch between "on" and "off" states very frequently. Considering the allele-level possibilities, at active genes, both alleles would be in an actively transcribing state. For PRCa genes, both alleles would be in an actively transcribing state, or both alleles would be in a silent PRC-marked state, or only one allele is in PRC-marked state, which, subsequently, would result in noisier gene expression. For PRC-repressed genes, both alleles are expected to be in a silent PRC-marked state

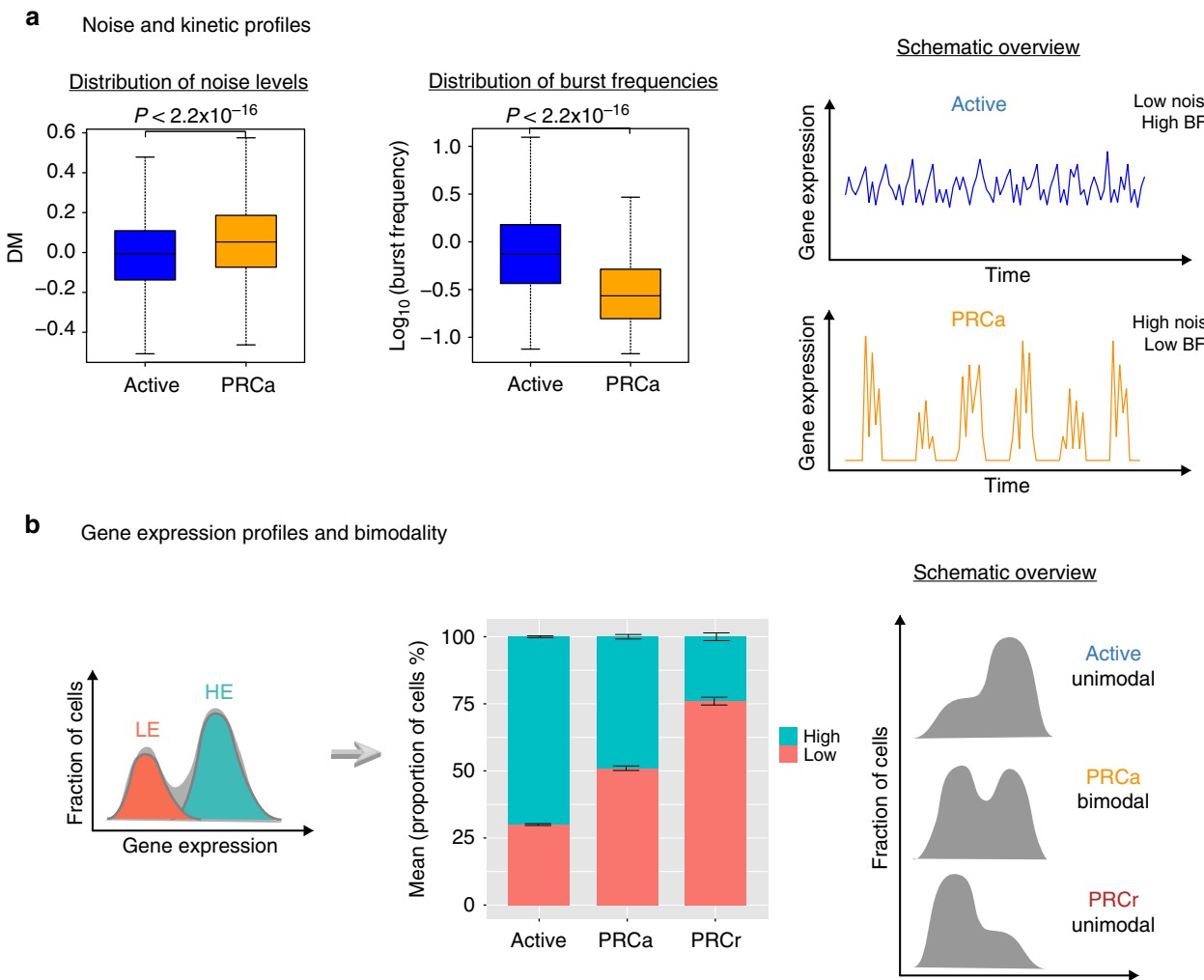

**Fig. 2** Stochastic gene expression of PRCa and active genes. **a** Comparison of PRCa and active genes reveals that PRCa genes are more variable with lower burst frequency levels than active genes ($P < 2.2 \times 10^{-16}$ by the two-tailed Wilcoxon rank sum test). Gene expression variation is represented by DM values. **b** Expression profiles of PRCa genes show bimodal patterns. The distribution of a gene with bimodal expression is assumed to be expressed as a mixture of two normal distributions (LE and HE states; *upper panel*). PRCa genes have mixed cell states (on average 49% in HE and 51% in LE) indicating they are either in active state (i.e., active RNAPII and no PRC marks) or in repressed state (unproductive RNAPII and with PRC marks) consistent with cellular heterogeneity, suggested in Brookes et al.[5] *Error bars* represent s.e.m.

To isolate pure stochastic gene expression from cell cycle variation in gene expression, we applied a latent variable model[13]. This is a two-step approach, which reconstructs cell cycle state before using this information to obtain "corrected" gene expression levels. The method reveals that the cell cycle contribution to variation is 1.2% on average (Supplementary Fig. 2B). While this effect is small, when clustering all cells based on G2/M stage markers, we found that cells separate into two groups: one with high expression of G2 and M genes and the other with low expression of these genes (Supplementary Fig. 2C). Applying the cell cycle correction removes this effect, leading to a more homogeneous expression distribution of these genes across the cells (Supplementary Fig. 2D).

To account for the technical noise present in single-cell RNA-seq data, we removed lowly expressed (LE) genes that are most likely to display high technical variability[14, 15]. Here, a gene is considered as LE if the average normalized read count is less than 10. This results in a set of 11,861 genes with moderate to high mRNA abundance. Subsequently, we use the DM (distance to median) to quantify gene expression variation in mRNA expression[11], since it accounts for confounding effects of

expression level and gene length on variation (described in detail in the Methods; Fig. 1).

Among the 11,861 expressed genes, 7175 have categorized ChIP-seq profiles as defined by Brookes et al.[5]; genes excluded have transcription start site (TSS) regions that overlap with other genes, and therefore cannot be unequivocally classified. We defined two major sets of genes based on their PRC marks and RNAPII states: (1) "Active" genes ($n = 4483$) without PRC marks (H3K27me3 or H2Aub1) but with active RNAPII (S5pS7pS2p), (2) "PRC-active" genes (labeled as "PRCa"; $n = 945$) with PRC marks (H3K27me3 or H3K27me3 plus H2Aub1), and active RNAPII.

To explore the transcriptional kinetics of these genes and describe stochastic gene expression in OS25 ES cells, we estimated their kinetic transcription parameters using a Poisson-beta model described previously[16] (see also in the Methods).

**PRCa genes have distinct transcriptional kinetics.** Using the DM measure to quantify gene expression variation in single cells, we observe that histone modifications mediated by PRCs

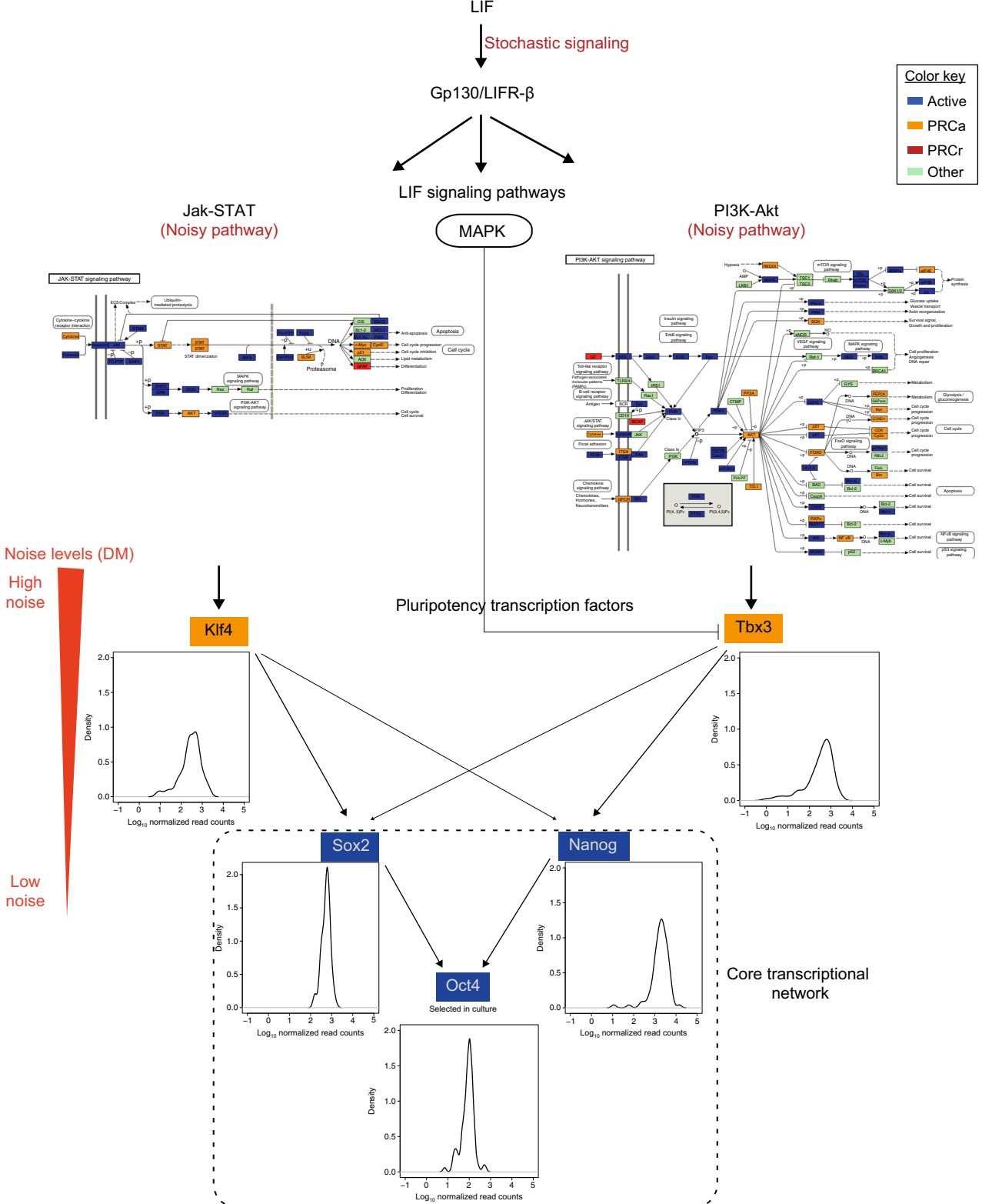

**Fig. 3** Signaling pathways that are key regulators of pluripotency in mESCs. In OS25 cells there is a selection for undifferentiated cells (high *Oct4*-expressing). LIF integrates signals into the core regulatory circuitry of pluripotency (*Sox2*, *Oct4*, and *Nanog*) via two signaling pathways: Jak-Stat and PI3K-Akt[22]. The Jak-Stat pathway activates *Klf4*, and the PI3K-Akt pathway stimulates the transcription of *Tbx3*, which are both PRCa genes. The MAPK pathway antagonizes the nuclear localization of *Tbx3*. PRCa genes are enriched in Jak-Stat and PI3K-Akt pathways, which show high cell-to-cell variation, suggesting a crucial role of PRCs in modulating fluctuations in signaling pathways that integrate LIF signals into core transcription factor network (Figure adapted from ref. [22])

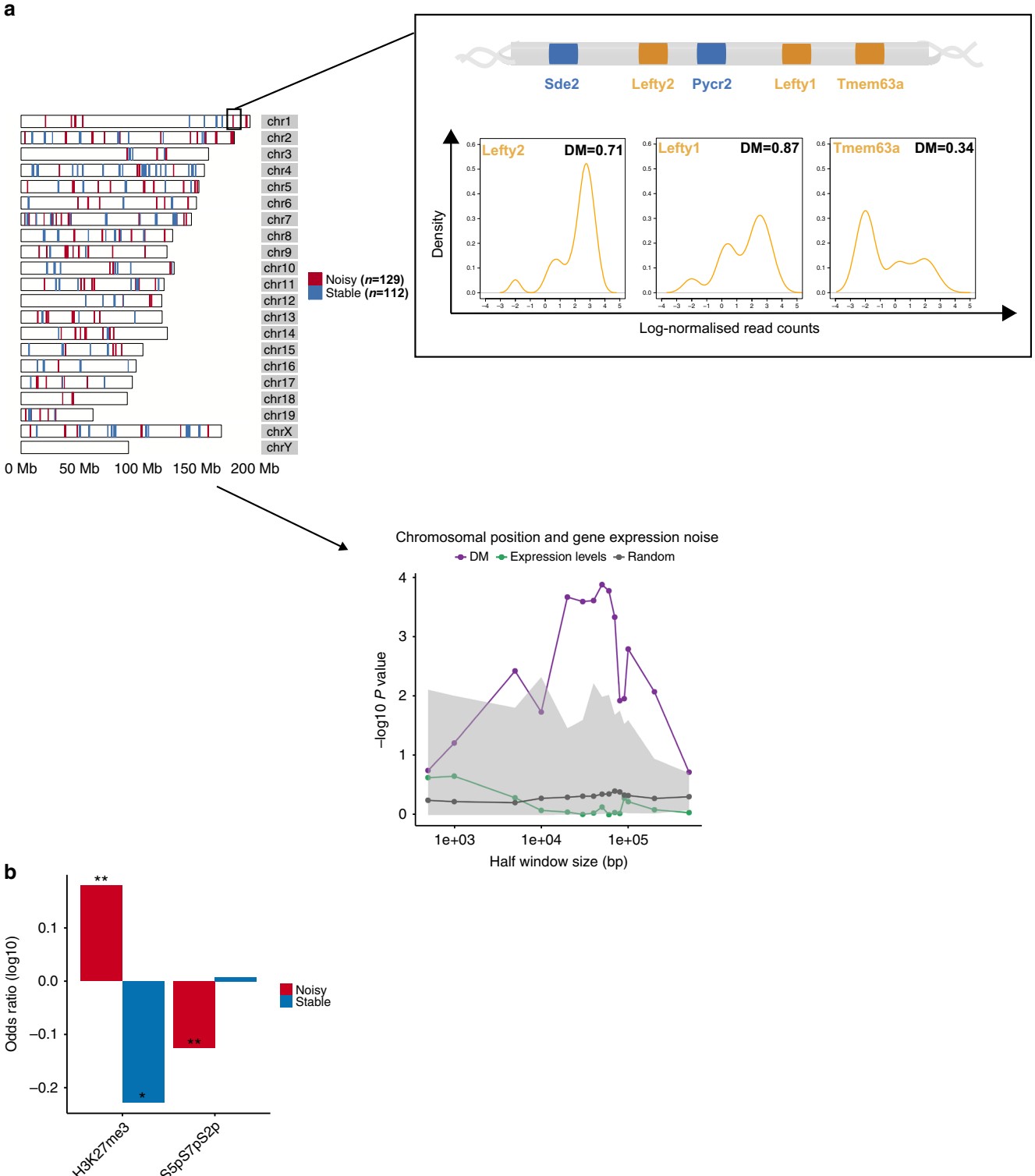

**Fig. 4** Chromosomal position effects and stochastic gene expression. **a** Maps of genes belonging to noisy and stable clusters. Chromosomal positions of genes marked by PRCs and RNAPII in the noisiest clusters. One of the noisy clusters is visualized, DM levels correlate with bimodal expression patterns. In *lower panel*, association between chromosomal position and gene expression noise is shown; the noise levels of genes in the neighborhood of noisy genes are significantly higher than those of flanking genes of stable genes. As a control, we constructed 100 randomized genomes in which the positions of genes were fixed but the DM value of each gene was assigned randomly without replacement, and the same analysis was performed on each randomized genome to obtain random *P* values. In all, 2.5% quantile of random *P* value, and 97.5% quantile of random *P* values are shaded in *gray*. All data are shown on a −log(p) scale. **b** Enrichments of PRC marks and RNAPII states in noisy and stable clusters; two-tailed Fisher's exact test; *$P < 0.1$, **$P < 0.05$

(H3K27me3 or H3K27me3 and H2Aub1) correlate with high levels of variability compared to active genes (those without PRC marks; $P < 2.2 \times 10^{-16}$ by the two-tailed Wilcoxon rank sum test, Fig. 2a). Furthermore, the inferred kinetic parameters provide insight into the expression behavior of genes, showing that active genes have significantly higher burst frequencies than PRCa genes (Fig. 2a and Supplementary Fig. 3A). This suggests that PRCa genes are more frequently in the "off" state, i.e., more alleles are in the off state at any given point in time, potentially due to the PRC repression of a subset of alleles.

To ensure that differences between the kinetic parameters are not driven by changes in gene expression levels between the active and PRCa groups, we extracted expression-matched genes of active and PRCa groups (please refer to Methods). These analyses confirmed that PRCa genes have lower burst frequency and higher noise levels than active genes (Supplementary Fig. 3B and C). Consequently, the greater cell-to-cell variability for PRCa compared to Active genes is not driven by difference in the mean expression level, but potentially linked to the presence of PRC marks themselves.

To explore whether H3K9me3 could contribute to the transcriptional heterogeneity identified at PRCa genes, we analyzed H3K9me3 ChIP-seq data of Mikkelsen et al.[17], and found that only a few expressed PRCa genes ($n = 5$) are marked by H3K9me3 at their promoter region (2 kb centered on the TSS), making further analysis statistically impossible.

Although the literature shows that the DNA of mESCs is hypomethylated, and genes that are marked by Polycomb are usually devoid of DNA methylation[18, 19], we checked the extent of DNA methylation at the PRCa gene list considered. We extracted the DNA methylation patterns at proximal promoter regions in mESCs reported in Fouse et al.[19]. Only a small proportion of genes ($n = 110$) has DNA methylation according to this definition. Owing to the small sample size, a statistical assessment will be weak, but comparison of gene expression variation profiles of these genes with the same number of PRCa genes (and same expression levels) that are unmethylated showed that the differences are not significant (Wilcoxon rank sum test, 0.1). This suggests no detectable effect of DNA methylation on transcriptional heterogeneity of PRCa genes (Supplementary Fig. 3D).

A decrease in the frequency of transcriptional bursting can manifest itself as a more bimodal pattern of gene expression across a cell population. Indeed, we observe that PRCa genes have significantly more bimodal expression profiles compared to active genes (see Methods for bimodality index calculation; Supplementary Fig. 3E and Fig. 2b). Assuming that the distribution of a gene with bimodal expression can be expressed as a mixture of two log-normal distributions[20] (LE and highly expressed (HE) states), we observe that PRCa genes have mixed cell states (on average 49% of cells in HE and 51% in LE). In contrast, active genes are mostly in the active state as expected (on average 70% in HE and 30% in LE). PRC-repressed genes with unproductive RNAPII and PRC marks, labeled as "PRCr") are 24% in HE and 76% in LE (Fig. 2b). Therefore, expression patterns of PRCa are in between Active and PRCr, suggesting a composite of these two states.

We should note that in our kinetic models, decay rates are set to 1 to normalize kinetic parameters so that they are independent of time[16]. To investigate whether decay rates have profound effects on kinetic parameters, we integrated published mRNA decay rates in mESCs[21] into our kinetic model. The subtle differences in decay rates across genes did not result in major changes in the inferred kinetic parameters, leaving all major trends unaffected (Supplementary Fig. 3F).

**PRCa genes are important regulators in signaling pathways**. To investigate potential functions of the cell-to-cell variation in gene expression in PRCa genes, we carried out KEGG pathway enrichment analysis for PRCa genes in our OS25 mESCs (see also Brookes et al.[5]). While active genes are enriched in pathways related to housekeeping functions, such as RNA transport, consistent with their uniform and stable expression across cells, PRCa genes are enriched in signaling pathways such as PI(3)K-Akt, Ras signaling, and TGF-beta signaling (Supplementary Table 1). These signaling pathways show high levels of cell-to-cell variation compared to pathways related to housekeeping functions (Supplementary Fig. 3G). This may be due to transcriptomic fluctuations introduced by cytokine LIF signalling via two signaling pathways: Jak-Stat3 and PI(3)K-Akt[22] (Fig. 3).

The Jak-Stat3 pathway activates *Klf4*, and the PI(3)K-Akt pathway stimulates the transcription of *Tbx3*[22]. The expression levels of *Klf4* and *Tbx3*, which are PRCa genes, are noisier than the pluripotency factors *Nanog*, *Sox2*, and *Oct4*. This pattern of noise propagation from the signaling pathways through the downstream transcriptional regulatory network is interesting, as it might indicate the role of PRCs in modulating transcriptomic fluctuations.

**Chromosomal position effects and stochastic gene expression**. It is known that neighboring genes on chromosomes exhibit significant correlations in gene expression abundance and regulation, partly due to two-dimensional (2D) chromatin domains[23–26]. Is there a similar effect of clustering by chromatin marks and noise in gene expression?

To address this, we investigated the positional effects of noise in mRNA expression using the DM values (Methods). If genes cluster together based upon their transcriptional noise, we would expect that the DM values of genes adjacent to noisy genes would be higher than those of genes adjacent to stable genes. Indeed, the noise levels of genes in the neighborhood of noisy genes are significantly higher than those of genes that flank stable genes ($P = 1.3 \times 10^{-4}$ by the one-tailed Wilcoxon rank sum test, ±50 kb of TSS, Supplementary Fig. 4A). This suggests that the genomic neighborhood might influence the frequency of transcriptional bursting.

In Fig. 4a, we show the association between chromosomal position and gene expression noise. The difference between the mean expression levels of flanking genes between noisy and stable genes is not significant ($P = 0.7311$ by the two-tailed Wilcoxon rank sum test, ±50 kb of TSS), suggesting that the clusters of genes are not driven by their expression levels. The association between chromosomal position and gene expression noise was most significant at the window size of 50 kb, but weaker at a neighborhood size of 0.5 Mb (Fig. 4a). (Please refer to Methods for $P$ value calculation.) Thus, genes tend to be clustered into neighborhood domains by their noise levels, ranging in size up to 0.5 Mb.

To identify the clusters of noisy or stable genes, we performed a sliding-window analysis on the mouse genome (Methods). We found 129 noisy clusters ranging in size from 4 to 11 genes, spanning a total number of 669 genes. Similarly, 112 stable clusters (between 4 and 13 genes) with a total number of 556 genes were found (Fig. 4a). The noise levels of genes in noisy clusters are significantly higher than that of genes in stable clusters ($P < 2.2 \times 10^{-16}$ by two-tailed Wilcoxon rank sum test, Supplementary Fig. 4B) independent of the mean expression levels and gene lengths (Supplementary Fig. 4C–D).

In addition, we found that DM levels correlate with bimodal expression patterns within the noisy clusters. One example is

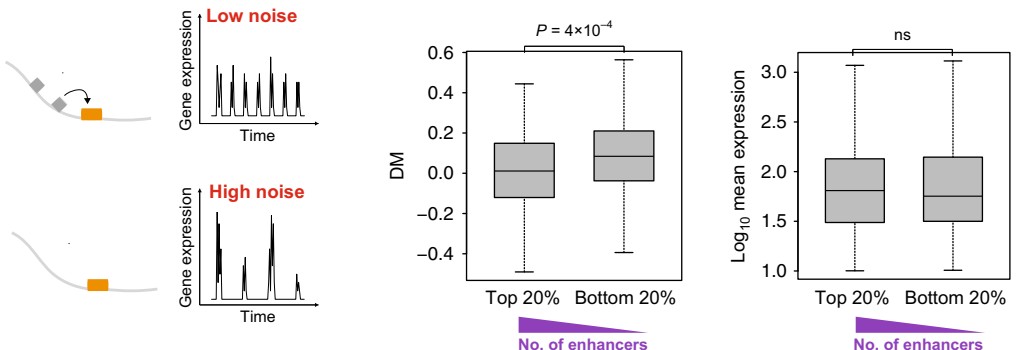

**Fig. 5** Effects of 2D and 3D neighborhood on transcriptional kinetics. **a** Histogram of simulated median distances under a null model assuming no positional preference in the neighborhood of PRCr genes. The observed median distance of PRCa genes to their nearest neighbor in the PRCr group, depicted by vertical dashed red line, are significantly less than expected by chance (empirical $P = 2 \times 10^{-2}$). **b** Analyzing mESC Promoter Capture Hi-C data reveals that PRCa genes have a strong enrichment for long-range contacts between promoters with levels in between PRCr and active genes. Error bars represent s.e.m. **c** PRCa genes have significantly more interactions with active enhancers compared to PRCr genes ($P < 2.2 \times 10^{-16}$). In contrast, interactions with poised enhancers are mainly observed for PRCr genes rather than PRCa ($P < 2.2 \times 10^{-16}$). **d** Occurrence of interactions with active enhancers decrease noise of PRCa genes independent of the mean expression levels

visualized in Fig. 4a; one of the noisy clusters on chromosome 1 consists of three PRCa and two active genes. *Lefty1* and *Lefty2* PRCa genes, which are important in controlling the balance between self-renewal and pluripotent differentiation in mESCs, are highly variable, and also highly correlated in their gene expression. An active gene, *Pycr2*, Pyrroline-5-carboxylate reductase 2, is in close proximity to both *Lefty1* and *Lefty2*, and is more variable than the *Sde2* gene that lies in proximity of *Lefty2* only (density profiles are shown in Supplementary Fig. 4E). Indeed, within the clusters, gene expression variation levels of active genes increase with the increasing number of flanking variable genes (Supplementary Fig. 4F). Another PRCa gene is *Tmem63a*, which is a transmembrane protein implicated in maintenance of pluripotency, lies near *Lefty1* and has high cell-to-cell variation in gene expression.

Interestingly, PRCs characterize the noisy clusters, i.e., PRC marks are enriched in noisy clusters rather than in stable ones. In particular, genes with H3K27me3 are enriched at noisy clusters ($P = 1.1 \times 10^{-2}$ by the two-tailed Fisher's exact test), but depleted at stable clusters ($P = 5.9 \times 10^{-2}$ by the two-tailed Fisher's exact test, Fig. 4b). Since PRCs are tightly associated with RNAPII states, we examined differences between the RNAPII state of genes between noisy and stable clusters. We found that genes marked by active elongating RNAPII (S5pS7pS2p) are depleted at noisy clusters ($P = 1.3 \times 10^{-3}$ by the two-tailed Fisher's exact test, Fig. 4b), supporting the view that elongating RNAPII modifications promote stable gene expression. Together, noisy clusters are characterized by the presence of PRC marks and the absence of active elongating RNAPII, while stable clusters are characterized by the absence of PRCs.

**Gene and enhancer clustering in 2D and 3D**. Next, we analyzed whether PRCa genes are proximal to fully repressed Polycomb genes, which could eventually increase their sensitivity to Polycomb repression. Linear spatial proximity between PRCa genes and PRCr genes is significantly closer than the median distance between randomly chosen genes and PRCr genes (empirical $P = 2 \times 10^{-2}$, Fig. 5a; Methods). Interestingly, PRCa genes are also in close proximity to active genes (empirical $P < 0.0001$, Supplementary Fig. 4G), while active genes are distal from PRCr genes (empirical $P = 5 \times 10^{-3}$, Supplementary Fig. 4H), suggesting a 2D spatial arrangement of these genes as Active-PRCa-PRCr (as visualized in Fig. 5a).

We next asked whether the linear genomic position effects of PRCs are reflected in the three-dimensional (3D) genome organization in ESCs. Recently, Schoenfelder et al.[27] found that PRC1 acts as a major regulator of ESC genome architecture by organizing genes into 3D interaction networks. They generated mESC Promoter Capture Hi-C (CHi-C) data[28], and analyzed it using the GOTHiC (Genome Organization Through Hi-C) Bioconductor package (http://www.bioconductor.org/packages/release/bioc/html/GOTHiC.html). This yielded a strong enrichment for long-range contacts between promoters bound by PRCs.

We applied the same approach to this data set using our gene list. We found that there is a strong enrichment for long-range promoter–promoter contacts for both PRCa and PRCr genes (Fig. 5b). Interestingly, PRCr genes have significantly stronger contact enrichment than PRCa genes in mESCs (one-tailed *t*-test $P = 6.3 \times 10^{-6}$). PRCa genes are in between PRCr and active genes; they have stronger contact enrichment than active genes (one-tailed *t*-test $P = 1 \times 10^{-4}$; Fig. 5b).

In Fig. 5b, the promoter contacts of the aforementioned noisy cluster PRCa gene *Lefty2* are visualized. It is in contact with the other PRCa genes *Lefty1* and *Tmem63a*, and it has a strong connectivity with the active *Pycr2* gene. These contacts may affect

*Pycr2*'s frequency of transcriptional bursting, and thereby tune expression noise.

In terms of the promoter preferences of gene sets, it is interesting to note that PRCa promoters interact equally with promoters of PRCr, PRCa, and active genes (Supplementary Fig. 4F). However, PRCr promoters have a distinct preference for other PRCr promoters (two-tailed Fisher's exact test $P < 2.2 \times 10^{-16}$).

We next investigated contacts between PRC promoter classes with putative regulatory (non-promoter) elements: enhancers. Enhancers are classified as in Schoenfelder et al.[27] as active (H3K4me1 and H3K27ac), intermediate (H3K4me1) or poised (H3K4me1 and H3K27me3). We found that PRCa genes have significantly more interactions with active enhancers compared to PRCr genes ($P < 2.2 \times 10^{-16}$; Fig. 5c). In contrast, interactions with poised enhancers are mainly observed for PRCr genes rather than for PRCa ($P < 2.2 \times 10^{-16}$).

Further, we asked whether interactions with enhancers affect transcriptional profiles of PRCa genes at the single-cell level. Interestingly, we found that interactions with active enhancers decrease noise in gene expression of PRCa genes. Sorting the PRCa genes based on the number of active enhancer interactions shows that more interactions lead to less noise in gene expression (two-tailed Wilcoxon rank sum test, $P = 4 \times 10^{-4}$). This stabilization of expression through active enhancers is independent of the mean expression levels (Fig. 5d).

In summary, these findings show that 3D genome architecture correlates with the chromatin state, and may influence noise in gene expression. This holds both in terms of promoter–promoter and enhancer–promoter interactions.

**Polycomb KO affects transcriptional profiles of PRCa genes**. To test whether noise in gene expression can be linked to Polycomb regulation mechanistically, we utilized conditional *Ring1B* double-knockout (dKO; in $Ring1A^{-/-}$ background) mES cells. These cells lack *Ring1A*, and have a tamoxifen-inducible conditional *Ring1B* deletion (Supplementary Fig. 5A and Methods). We confirmed *Ring1B* deletion 48 h post-tamoxifen induction, and generated single-cell RNA-seq data for both untreated (*Ring1A* single KO) and tamoxifen-treated dKO (*Ring1A* and *Ring1B*) mES cells (see Methods). In these conditions, RING1B protein is lost in ~ 48 h, and H2Aub1 modification is no longer detected on chromatin and Polycomb-repressed genes are derepressed without loss of pluripotency factors *Nanog*, *Oct4*, and *Rex1* (refs 5, 8, 29).

We compared the changes in the mean expression at PRCr, PRCa, and active genes. We found that PRCr shows substantial derepression after *Ring1A/B* dKO (Fig. 6), as expected from bulk mRNA-seq and microarray data[5, 29]. The mean expression change at PRCa genes is lower than at PRCr ($P = 4.1 \times 10^{-9}$ by the two-tailed Wilcoxon rank sum test; Fig. 6) more likely due to the fact that they are already expressed to some extent in untreated cells. Nevertheless, changes in the mean expression at PRCa genes are higher than those at active genes ($P = 2 \times 10^{-7}$ by the two-tailed Wilcoxon rank sum test; Fig. 6 and Supplementary Fig. 5B). Increased expression of PRCa genes upon *Ring1A/B* dKO recapitulates previous findings using bulk transcriptomic analyses[5, 8].

Importantly, comparison of noise levels shows that there is a more pronounced decrease in noise levels at PRCa genes compared to active genes upon *Ring1A/B* dKO ($P = 4 \times 10^{-3}$ by one-tailed *t*-test; Supplementary Fig. 5C). This supports our findings that Polycomb tunes gene expression noise. In addition, there is a more pronounced decrease in bimodality at PRCa genes (Supplementary Fig. 5D), while burst frequency levels

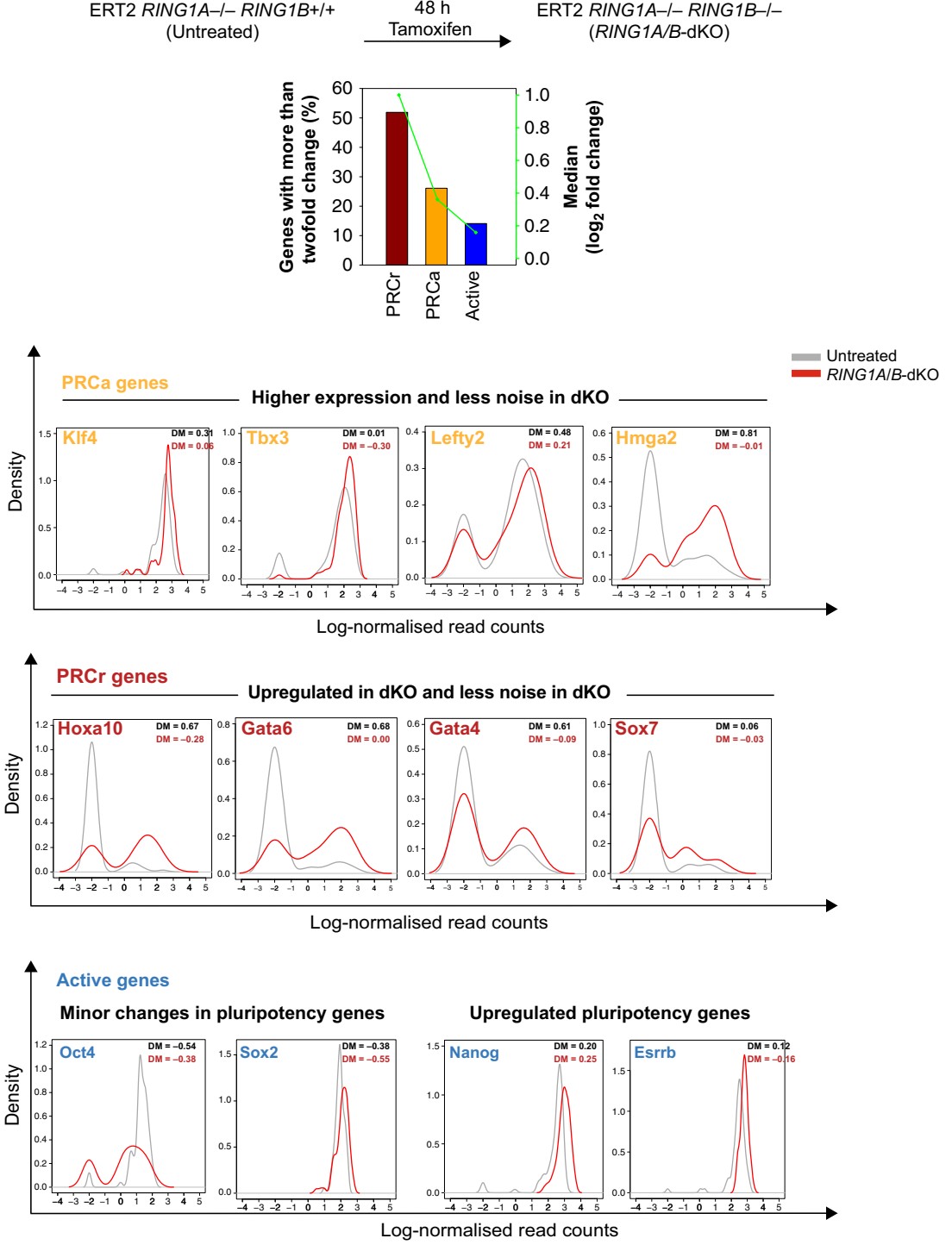

**Fig. 6** Single-cell profiling of *Ring1A/B* dKO mES cells'. PRCr show substantial derepression after *Ring1A/B* dKO. The mean expression change at PRCa genes is lower than that at PRCr and, in contrast, is higher than that at active genes. Comparison of noise levels shows that there is a more pronounced decrease in noise levels at PRCa genes compared to active genes. Gene expression profiles of some important genes for ESC biology are shown. Key pluripotency transcription factors *Klf4* and *Tbx3* are more expressed and less noisy in *Ring1A/B* dKO cells. Other transcriptional regulators such as *Hmga2* and *Hdac2* become upregulated after dKO. Consistently, key differentiation markers such as *Gata4*, *Gata6* are upregulated. Among active pluripotency factors, *Oct4* and *Sox2* show minor changes in expression (mean expression levels are not significantly different), and *Nanog* and *Esrrb* are upregulated

decrease more significantly at active genes (Supplementary Fig. 5E).

Among PRCa genes, key pluripotency transcription factors *Klf4* and *Tbx3* and other transcriptional regulators (such as *Hmga2* and *Hdac2*) important for ESC biology become upregulated and show less noisy profiles after *Ring1A/B* dKO

(gene expression profiles are shown in Fig. 6). In addition, key differentiation markers such as *Gata4*, *Gata6*, which are PRCr genes, are upregulated upon dKO (Fig. 6), implying that a Polycomb KO could make cells more prone to differentiation (as expected from refs [5, 29]). The same pattern of differential gene expression is also observed in the bulk RNA-seq data. Taken

together, these findings indicate the key role of Polycomb in regulating transcriptional profiles of PRC-bound genes.

We observe that non-PRC targets (i.e., active genes) show subtle trends in change in gene expression; expression levels of active pluripotency factors such as *Oct4* and *Sox2* show minor changes in gene expression. In contrast, *Nanog* and *Esrrb* are upregulated (Fig. 6), suggesting that Polycomb may indirectly control the expression of genes specifically associated with pluripotency. Expression patterns of all these genes can be found at http://www.ebi.ac.uk/teichmann-srv/espresso/.

## Discussion

It is well understood how post-translational modifications of histones, including acetylation, methylation, phosphorylation, and ubiquitination, modulate the chromatin structure, thereby affecting the regulation of gene expression levels[30]. It is much less well understood how chromatin status is related to the kinetics of transcription in terms of transcriptional bursting. Differences in stochastic gene expression lead to different degrees of cell-to-cell variation in expression levels, even for genes with the same mean expression across an ensemble of cells. Recent molecular studies have shown that individual cells can show substantial differences in both gene expression and phenotypic output[31, 32]. Genetically identical cells may still behave differently under identical conditions[33]. This non-genetic variability is mainly due to cell-to-cell variation in gene expression[34, 35], which relates to each gene's chromatin status[36]. Noisy or stochastic gene expression profiles may play an important role in the regulation of ES cells[37].

In this work, we focus on histone modifications that are mediated by PRCs and investigate their relationship with stochastic gene expression in mESCs. Earlier work indicated that expression of Polycomb target genes negatively correlates with levels of H3K27me3, and suggested that dynamic fluctuations in chromatin state are associated with expression of certain Polycomb targets in pluripotent stem cells[38]. Although PRCs are known to exert a repressive effect, interestingly, the cohort of PRC-bound genes contains not only silent genes, but also genes with intermediate and high expression[5]. A large range of expression levels at PRC-target genes is observed in published mRNA data sets[5, 39] and substantial expression has been previously observed at PRC2-target genes[40, 41]. The moderate to high expression levels at some PRC-bound genes allow us to reliably quantify gene expression variation (which is not possible if expression is too low).

Here, benefiting from the power of single-cell RNA-seq analysis, we show that PRCa genes have greater cell-to-cell variation in expression than their non-PRC counterparts, suggesting that they switch between on and off states in a more dramatic way. Along the same lines, their expression patterns are more likely to be bimodal, suggesting a composite of active and PRC-repressed states at the single-cell level. These findings indicate the role of Polycomb in modulating frequency of transcriptional bursting and thereby tuning gene expression noise.

Transcriptional bursts that arise from random fluctuations between open and closed chromatin states of a gene are one of the major sources of gene expression noise in eukaryotes[42]. Since these fluctuations are modulated by transcription factors, nucleosomes and chromatin remodeling enzymes, we can speculate that gene expression noise may be linked to chromosomal position through shared chromatin domains with specific characteristics such as histone modifications. Consistent with this notion, several studies using a reporter transgene integrated in multiple loci have shown that gene expression noise varies with chromosomal position in yeast and mammalian cells[43–48].

However, large-scale studies measuring noise in protein expression of endogenous genes could not find a strong association between chromosomal position and gene expression noise in yeast[34, 49]. This discrepancy might be due to gene-specific confounding factors and different statistics to examine the association. For example, essential genes with low noise derived from the same data sets of the large-scale proteomic studies are clustered into neighborhood domains with low nucleosome occupancy[23]. More importantly, noise in protein expression is not a good measure for examining the effect of chromatin regulation on transcriptional bursting since slowly degrading proteins can buffer transcriptional noise at the protein levels[46]. Given the lack of high-throughput measurements of noise in mRNA expression of endogenous genes in eukaryotes, it is not clear whether genes are distributed across the genome by their noise levels and which chromatin features modulate the chromosomal position effects.

Analysis of the chromosomal position effects of noise reveals that genes are significantly clustered according to their noise levels, which are mainly modulated by PRCs. Interestingly, across the chromosomes, we found that PRCa genes are in close proximity to fully repressed PRC targets. This could increase their sensitivity to PRC repression, and explain their ability to switch between active and repressed states in a more dramatic way than other genes.

In addition to 2D spatial proximity of genes, long-range regulatory interactions have a key role in gene expression control[50]. Recently, analyzing mESC Promoter Capture Hi-C (CHi-C) data[28], Schoenfelder et al. showed that PRC1 acts as a major regulator of the ESC 3D genome architecture[27]. Applying the same methodology, we show that there is a strong enrichment for long-range promoter–promoter contacts for both PRCa and PRCr genes. Interestingly, interactions with active enhancers decrease gene expression noise (but not the mean expression levels) of PRCa genes, suggesting that 3D genome architecture has a key role in controlling gene expression noise.

To further decipher the role of PRCs in regulating gene expression and noise, we performed single-cell RNA-seq for both PRC expression (*Ring1A*-KO, untreated) and PRC-deleted (*Ring1A/B*-dKO, tamoxifen-treated) mESCs. We observe substantial derepression of PRC-bound genes after *Ring1A/B*-dKO as expected. The mean expression changes at PRCa genes are significantly lower than those at PRCr genes, supporting the fact that they are already expressed in untreated ES cells. Moreover, in terms of noise profiles, we observe a significant decrease in noise levels of PRCa genes compared to active genes. This genetic validation supports our findings that Polycomb plays a key role in modulating the kinetics of stochastic gene expression.

## Methods

**Single-cell RNA-seq of mouse OS25 ESCs**. Mouse ES-OS25 cells were grown on 0.1% gelatin-coated surfaces in supplemented GMEM-BHK21 containing 10% fetal calf serum (FCS, PAA Laboratories, Gmbh), non-essential amino acids, sodium pyruvate (1 mM), sodium bicarbonate (0.075%), streptomycin (100 units/ml), L-glutamine (292 μg/ml), penicillin G (100 units/ml), 2-mercaptoethanol (0.1 mM), and human recombinant LIF (1000 units/ml, Chemicon)[8, 51].

For single-cell sequencing libraries were prepared according to Fluidigm manual "Using the C1 Single-Cell Auto Prep System to Generate mRNA from Single Cells and Libraries for Sequencing". OS25 cell suspension was loaded on 10–17 micron C1 Single-Cell Auto Prep IFC, Fluidigm; cDNA was synthesized in the chip using Clontech SMARTer Kit, and Illumina sequencing libraries were prepared with Nextera XT Kit and Nextera Index Kit (Illumina). Libraries from 96 cells were pooled and sequenced on four lanes on Illumina HiSeq2000 using 100 bp paired-end protocol.

**Mapping Reads**. For each cell, paired-end reads were mapped to the *M. musculus* genome (GRCm38) using Genomic Short-read Nucleotide Alignment Program (GSNAP) with default parameters[52]. Next, uniquely mapped reads to the genome

were counted using htseq-count (http://www-huber.embl.de/users/anders/HTSeq/) and normalized with size factors using DESeq[53].

**Classification of genes based on ChIP-seq profiles**. To integrate ChIP-seq data with single-cell RNA-seq, we mapped 18,860 UCSC known gene IDs from Brookes et al.[5] to Ensembl IDs using BioMart[54]. Then, we categorized the genes based on Brookes et al. classification: (1) "Active" genes ($n = 4732$) are defined as those without PRC marks (H3K27me3 or H2Aub1) but with active RNAPII (S5pS7pS2p), (2) "PRCa" ($n = 1263$) genes are marked by PRCs (H3K27me3 or H3K27me3 plus H2Aub1) and active RNAPII, (3) "PRCr" genes ($n = 954$) have both PRC marks (H3K27me3 and H2Aub1), unproductive RNAPII (S5p only and not recognized by antibody 8WG16) and not expressed in bulk mRNA data by Brookes et al. (bulk mRNA FPKM <1). We should note that vast majority of PRCa and PRCr genes are H3K4me3-positive (1248 out of 1263 PRCa, and 938 out of 954 PRCr; see Brookes et al.[5])

We focus on Active and PRCa genes with moderate to high mRNA abundance and, therefore, we remove genes that have mean normalized counts lower than 10. Thus, in the final gene set, there are 4483 active genes and 945 PRCa genes.

For H3K9me3, reads from Mikkelsen et al.[17] were mapped to mouse genome (mm9, July 2007) using Bowtie2 v2.0.5[55], with default parameters. Enriched regions were identified with BCP v1.1[56] in Histone Mark mode, using as control H3 from Mikkelsen et al., processed in the same way. Genes were defined as positive for H3K9me3 at their promoter or gene body when an enriched region was overlapping with a 2 kb window around the TSS or between the TSS and TES, respectively.

**Inference of transcriptional kinetic parameters**. To explore kinetics of stochastic gene expression, we fitted a Poisson-beta model[16]. Poisson-beta model is an efficient way to describe the long-tailed behavior of mRNA distribution resulting from occasional transcriptional bursts as well as to explain expression bimodality of genes with low burst frequency. Transcriptional kinetic parameters are characterized by two parameters, burst size is described as the average number of synthesized mRNA molecules while a gene remains in an active state and burst frequency is the frequency at which bursts occur per unit time. To ensure that the parameters are statistically identifiable, the parametric bootstrap for goodness-of-fit testing is used as applied in ref. [16]. Out of 5428 genes (active and PRCa), 4526 genes (83%) have identifiable estimates of kinetic parameters. We focus henceforth on these genes in analysis of burst size and frequency.

We should note that our kinetic analyses do not account for technical noise as our data do not contain external spike-in molecules (the only way to incorporate technical noise in our kinetic model). Therefore, we addressed this point by focusing on moderately or highly expressed genes with an expression cutoff of 10. The assumption is that technical noise for these genes is small enough to estimate kinetic parameters accurately. We should also note that our results are robust to changes in selection of expression cutoff (Supplementary Fig. 3A).

**Controlling for expression levels in kinetic models**. To control for expression levels for PRCa and active gene sets, we extracted expression-matched sets of active and PRCa genes using "matching" function in R "arm" package with default settings. In this way, an active gene is matched to a PRCa gene that has the closest mean expression level.

**Calculating DM as a measure of gene expression variability**. Widely used measures for quantifying gene expression variation in mRNA expression such as the coefficient of variation (CV) and Fano factor are not suitable for assessing differences in gene expression variation between genes because they depend strongly on gene expression levels and gene length. To properly account for the confounding effects of expression level and gene length on variation, we first computed a mean corrected residual of variation by calculating the difference between the observed squared CV (log10-transformed) of a gene and its expected squared CV. As a second step to correct for the effect of gene length on the mean corrected residual of variation, we calculated the difference between the mean corrected residual of the gene and its expected residual, which is referred to as DM[11]. The expected squared CV or the expected residual was approximated by using a running median.

**Calculation of bimodality index**. Bimodality index is a measure to identify and rank bimodal signatures from gene expression data, and was calculated according to definition introduced by Wang et al.[20]. The distribution of a gene with bimodal expression is assumed to be described as a mixture of two normal distributions with equal standard deviation. Proportions of observations in two components were estimated using R package "mclust".

**Identifying noisy and stable genes across mouse chromosomes**. To investigate the position effects of noise in mRNA expression using DM values, we first sorted all expressed genes ($n = 11,861$) in descending the order according to their DM values and chose the top 20% as "noisy" genes and the bottom 20% as "stable" genes. For each gene, we counted the number of noisy (or stable) genes (excluding

the focal gene) in the neighborhood of the gene ($\pm 0.5$ kb ~ 500 bp of the TSS of the focal gene).

While investigating the association between chromosomal position and gene expression noise, as a control, we constructed 100 randomized genomes in which the positions of genes were fixed but the DM value of each gene was assigned randomly without replacement, and the same analysis was performed on each randomized genome. The P values observed in the real genome are less than the median of P values found in the randomized genomes at all neighborhood sizes and even less than the 2.5% quantile of random P values at the neighborhood sizes between 20 kb and 0.2 Mb (Fig. 4a).

**Identifying clusters of genes by a sliding-window approach**. To identify the clusters of noisy or stable genes in the mouse genome, we used a sliding-window approach[57] with a window size of four genes. Given a set of genes having valid DM values, a window starts from the first gene of each chromosome and keeps shifting right by one gene until it reaches the end of the chromosome. We ignored windows having a distance between TSSs of the first and fourth genes of the windows larger than (window size—1) × 0.5 Mb. We measured the overall noise of each window by summing rolling means of the DM values of two consecutive genes within the window. We then calculated this noise score of randomly chosen four genes, and repeated this process 100,000 times, yielding a null distribution of the overall noise score of a window. We called a window to be significantly noisy (or stable) if its noise score is above 97.5% of randomized windows (or below 2.5% of randomized windows). Finally, we merged all overlapped noisy (or stable) windows to construct a set of noisy (or stable) clusters.

The total number of genes in noisy clusters found in the mouse genome is not significantly higher than that of 1000 randomized genomes (empirical $P = 0.3996$). In contrast, the total number of genes in stable clusters is significantly lower than expected by chance (empirical $P = 1.0 \times 10^{-3}$), suggesting that the stable clusters are relatively rare.

**Testing the spatial proximity between PRCa and PRCr genes**. To test whether PRCa genes are in the neighborhood of PRCr genes, we calculated the distance for each gene in the PRCa group (1263 genes) to its nearest neighbor in the PRCr group (954 genes) using TSSs. The observed mean and median distance were tested against a null model, assuming no positional preference of PRCa genes in the neighborhood of PRCr genes. We observed that a majority of genes not expressed in mESCs are distal from Active, PRCa, and PRCr genes. To correct for the effect of these inactive genes, we defined a background set of genes as ones belonging to Active, PRCa, or PRCr genes. We randomly sampled 1263 genes from the background set by excluding genes that are in the PRCr group or in the chromosomes on which the 954 PRCr genes are not located, and calculated the mean and median distance between the randomly chosen genes and PRCr genes. We repeated this process 10,000 times and computed the empirical P values of the observed mean and median distance based on a null distribution of simulated distances.

**Promoter–promoter contacts and contact enrichment analysis**. Significant promoter–promoter and promoter–genome interactions in Wild Type (WT) ESC were obtained from Schoenfelder et al.[27]. Short-range intrachromosomal contacts were excluded by filtering contacts separated by <10 Mb. To measure the enrichment of contacts within a set of promoters, 100 random promoter sets were generated with comparable pair-wise distance distributions to the experimental set. Contact enrichment was derived by dividing the number of contacts in the experimental set by the average number of contacts in the control sets. For each experimental set, we calculated the contact enrichment using three independent control sets and showed the mean contact enrichment and the standard deviation. Contact enrichment differences were evaluated using one-tailed t-tests.

**Gene ontology and KEGG pathway analyses**. Annotation of KEGG pathways[58] and their associated genes was retrieved using Bioconductor Package KEGGREST. Enrichment of KEGG pathways was assessed by Fisher's exact test in R Stats package, and P values were adjusted for multiple testing by calculating false discovery rates.

**Ring1A/B dKO cells and mRNA sequencing**. Ring1A/B dKO cells[29] (a kind gift from Neil Brockdorff, which have been authenticated before) with constitutive Ring1A KO and tamoxifen-inducible conditional Ring1B KO were cultured on mitomycin-inactivated feeders in DMEM (lacking pyruvate; Gibco), supplemented with 10% batch-tested FCS (Labtech), 50 mM ß-mercaptoethanol, L-glutamine (Gibco), sodium pyruvate (Gibco), non-essential amino acids (Gibco), penicillin/streptomycin (Gibco) supplemented with 1000 U/ml LIF (Milipore)[29]. These cell lines have been tested and were found to have no mycoplasma contamination. Feeders in Ring1A/B dKO mES cells (untreated and tamoxifen-treated) are depleted using Feeder removal MicroBeads (Miltenyi Biotec). To induce Ring1b KO, cells are cultured in media containing 800 nM 4-hydroxytamoxifen (Sigma) for 48 h and confirmed using genomic DNA isolation and PCR across Cre-excised region[8, 29]. Primer information[29] is listed below.

Ring1b-s3      AAGCCAAAATTTAAAAGCACTGT

*Ring1b*-4681 ATGGTCAAGCAAACATGAAGGT
*Ring1b*-as4 TGAAAAGGAAATGCAATGGTAT.

All cells are processed on C1 Single Cell Auto Prep System (Fluidigm; 100–7000 and 100–6209) using medium sized C1 mRNA-seq chips (10–17 µm; 100–5670) with ERCC spike-ins (Ambion; AM1780) following the manufacturer's protocol (100–5950 B1) requiring SMARTer kit for Illumina Sequencing (Clonetech; 634936). Single-cell libraries were made using Illumina Nextera XT DNA sample preparation kit (Illumina; FC-131-1096) after cleanup and pooling using AMPure XP beads (Agencourt Biosciences; A63880). Each library is sequenced on single HiSeq2000 lane (Illumina) using 100 bp paired-end sequencing.

We also generated standard bulk RNA-seq for each condition. Bulk RNA-seq libraries were prepared and sequenced using the Wellcome Trust Sanger Institute sample preparation pipeline with Illumina's TruSeq RNA Sample Preparation v2 Kit. RNA was extracted from 1 to 2 million cells using the QIAGEN RNA Purification Kit on a QiaCube robot. The quality of the RNA sample was checked using gel electrophoresis. For library preparation, poly-A RNA was purified from total RNA using oligo-dT magnetic pull-down. Next, mRNA was fragmented using metal-ion-catalyzed hydrolysis. The cDNA was synthesized using random hexamer priming, and end repair was performed to obtain blunt ends. A-tailing was done to enable subsequent ligation of Illumina paired-end sequencing adapters, and samples were multiplexed. The resulting library was amplified using 10 cycles of PCR. Samples were diluted to 4 nM, and 100 bp paired-end sequencing was carried out on an Illumina HiSeq2000. Quality control of sequencing was performed by the Sanger sequencing facility.

We observed that average single-cell expression levels recapitulated the bulk gene expression levels with a Spearman rank correlation coefficient of 0.89 and 0.88 for untreated and dKO conditions, respectively.

**Data availability**. Sequencing data are available in the ArrayExpress database (http://www.ebi.ac.uk/arrayexpress) under accession number E-MTAB-5661. All other data are available from the authors upon reasonable request.

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

## Acknowledgements

We are grateful to Neil Brockdorff for the kind gift of *Ring1A/B* dKO cells. We acknowledge funding from the Wellcome Trust Strategic Award on Mouse Gastrulation, the Lister Institute, and the Helmholtz Association. E.T.T. acknowledges an EMBO short-term fellowship (ASTF 336-2015). J.K.K. acknowledges DGIST Start-up Fund (2017010073) of the Ministry of Science, ICT and Future Planning.

## Author contributions

G.K.: Carried out statistical and bioinformatics analyses, interpreted data, and prepared figures and the manuscript; J.K.K.: Applied methods for deciphering chromosomal position effects of noise and contributed to figure and manuscript preparation; A.A.K.: Performed single-cell RNA-seq experiments of OS25 mESCs; E.T.T. Performed single-cell RNA-seq experiments of OS25 mESCs and contributed to manuscript preparation; K.N.N.: Performed single-cell RNA-seq experiments of *Ring1A/B* dKO cells; B. M.: Carried out promoter contact enrichment analysis; S.E. and J.C.M.: Advised on analysis; A.P.: Advised on cell culture conditions and ESC biology and contributed to manuscript preparation; S.A.T.: Designed experiments, advised on analysis, and contributed to manuscript preparation.

## Additional information

**Competing interests:** The authors declare no competing financial interests.

