## [Peer Review File · Nature Communications]

Reviewers' comments:

Reviewer #1 (Remarks to the Author):

The authors have addressed most of the questions that can be supported by informatics. There are several points that the authors rebutted to be outside the scope of the paper. This is unfortunate, as they can potentially lead to new insights or solidify the specific claims.

Overall, the work is rather solid. However, the conclusions are not exciting or particularly novel. Hence, it is more suitable for a specialized journal and not for Nature Communications.

Reviewer #2 (Remarks to the Author):

The authors have addressed all of my concerns and it should be accepted for publication now.

Reviewer #3 (Remarks to the Author):

The revised manuscript does not address my core concerns, and contains a number of incorrect and unsupported statements. I repeat my major concerns below:

1. The response claim that this paper is the ' first transcriptome-wide study showing that differences in chromatin state can affect gene expression kinetics', and that none of the previous papers addressed the mechanistic nature of expression variation. These statements are incorrect, as has also been pointed out by the other reviewers. Both Kumar and Kolodziejczyk -- in the same system -- link expression heterogeneity to changes in chromatin state (in the latter case, by showing that differences exist across serum conditions, which have widely been linked to changes in H3K27me3 levels). The former manuscript ties changes in heterogeneity to miRNA-production, and explicitly tests the relationship between H3K27me3 and expression heterogeneity.

2. I continue to find the kinetic analyses to be fundamentally flawed. The core of my argument is that the kinetic model used does not incorporate or account for technical noise in any way. The revision is clear that there are 20-40% technical noise, therefore the parameters inferred from this model are not meaningful.

While the response states that the authors 'do not change', by changing the cutoff from 10 reads to 100 reads, it is clear that there is a significant shift. Looking at the change in burst frequency in the original manuscript, there is an enormous shift with a highly significant p-value. Moving this cutoff (corresponding to a reduction in technical noise of 40-20%), the distributions appear nearly overlapping, and the p-value is now >0.01 . To suggest that these results are unchanged is misleading and inaccurate - a key result (Figure 1A) of the manuscript is heavily dependent on the selection of an improper cutoff, and is flawed.

To reiterate, the choice of 10 reads/cell is an inappropriate, cutoff, since when taking the total reads per cell into account, this represents ~ 1 molecule per cell. This can hardly be considered to be a moderately to highly expressed gene. Even multiplying this by 10 does not reach a threshold that can be considered to be immune from technical noise in scRNA-seq experiments.

3. The response states that RNA FISH and single cell RNA-seq results consistently agree, and therefore there is no need for external validation. This is again an inaccurate statement. Numerous studies have described the extensive technical biases introduced by amplification of single cell RNA-seq data, which could explain the results the authors see here, but are not present in RNA FISH.

The authors cite Grun et al., to suggest that the methods agree. Grun et al of course shows that single cell RNA-seq data is far less sensitive than RNA-FISH. However, more importantly, the data in Grun et al. contained unique molecular identifiers, which control for bias in PCR amplification, enabling them to match characteristics of the two distributions. The data here do not include UMIs, and so the comparison is both invalid and inaccurate.

Response to Reviewer #3:

We thank Rev. #3 for his/her comments, which enabled us to clarify important points in our manuscript. The remarks of Rev. #3 are denoted in italics.

The revised manuscript does not address my core concerns, and contains a number of incorrect and unsupported statements. I repeat my major concerns below:

1. The response claim that this paper is the 'first transcriptome-wide study

showing that differences in chromatin state can affect gene expression kinetics', and that none of the previous papers addressed the mechanistic nature of expression variation. These statements are incorrect, as has also been pointed out by the other reviewers. Both Kumar and Kolodziejczyk -- in the same system -- link expression heterogeneity to changes in chromatin state (in the latter case, by showing that differences exist across serum conditions, which have widely been linked to changes in H3K27me3 levels). The former manuscript ties changes in heterogeneity to miRNA-production, and explicitly tests the relationship between H3K27me3 and expression heterogeneity.

The reviewer's statement about our earlier work (Kolodziejczyk et al. *Cell Stem Cell* 2015) is incorrect; this study has addressed the roles of pluripotency and cell cycle genes in mESCs across different conditions, however, histone modifications or H3K27me3 levels are not studied explicitly in any part of this study.

It is true that there may be implicit information in an indirect sense via cell culture conditions in these previous papers, but such global relationships do not provide a direct molecular insight as shown in our work here. To be specific, while Kumar et al. addressed the relationship between H3K27me3 levels and expression heterogeneity, this paper or previous papers do not focus on the mechanistic role of Polycomb in regulating transcriptional bursting. Furthermore, Kumar et al. provided an analysis of serum/LIF-cultured cells, and we have shown that these cells are a mixture of multiple distinct biological states (Kolodziejczyk et al., *Cell Stem Cell*, 2015). This makes these samples unsuitable to study transcriptional bursting, as much of the variation in gene expression will be due to cell subpopulations, rather than *bona fide* transcriptional bursting that can be ascribed to chromatin state.

The LIF conditions under Oct4 selection that we focus on in this manuscript are in this sense ideal. With the present study, we do not wish to address the differences between naïve and less naïve states of differentiation, and instead our work here is focused on Polycomb repression and its impact

on transcriptional cell-to-cell variation. Our paper is conceptually entirely distinct from the previous papers cited in this sense.

We had already described this in the previous revised manuscript. Now, in the revised version, we cite Kumar et al. in the discussion part of the manuscript as well, and have modified the text as follows:

“Earlier work indicated that expression of Polycomb target genes negatively correlates with levels of H3K27me3, and suggested that dynamic fluctuations in chromatin state are associated with expression of certain Polycomb targets in pluripotent stem cells³⁸.”

Additionally, we have now modified the last paragraph of the introduction as follows:

“Motivated by this hypothesis, we integrate states of histone and RNAPII modification from a published classification of ChIP-Seq data⁵ with single-cell RNA-sequencing data generated for this analysis. The matched chromatin and scRNA-seq data sets allow us to decipher, on a genome-wide scale, how differences in chromatin state affect transcriptional kinetics.”

2. I continue to find the kinetic analyses to be fundamentally flawed. The core of my argument is that the kinetic model used does not incorporate or account for technical noise in any way. The revision is clear that there are 20-40% technical noise, therefore the parameters inferred from this model are not meaningful.

The reviewer is right that our kinetic analysis does not account for technical noise, as we have already explained this in the previous rebuttal letter:

“Since our data do not contain external spike-in molecules (the only way to incorporate technical noise in our kinetic model), we addressed this point by focusing on moderately or highly expressed genes with an expression cutoff of 10. The assumption is that technical noise for these genes is small enough to estimate kinetic parameters accurately.”

Additionally, in the previous rebuttal letter, we have also shown that the fraction of technical noise saturates to 20% at the mean normalised count of 100.

Now, to make this point clearer, in the revised manuscript, we have also explained this in the

Methods section of the manuscript as follows:

“We should note that our kinetic analyses do not account for technical noise as our data do not contain external spike-in molecules (the only way to incorporate technical noise in our kinetic model). Therefore, we addressed this point by focusing on moderately or highly expressed genes with an expression cutoff of 10. The assumption is that technical noise for these genes is small enough to estimate kinetic parameters accurately.”

While the response states that the authors 'do not change', by changing the cutoff from 10 reads to 100 reads, it is clear that there is a significant shift. Looking at the change in burst frequency in the original manuscript, there is an enormous shift with a highly significant p-value. Moving this cutoff (corresponding to a reduction in technical noise of 40-20%), the distributions appear nearly overlapping, and the p-value is now >0.01. To suggest that these results are unchanged is misleading and inaccurate - a key result (Figure 1A) of the manuscript is heavily dependent on the selection of an improper cutoff, and is flawed.

In the previous rebuttal letter, we have shown distributions of noise and burst frequency levels with a stricter cutoff of 100 mean normalized counts. However, we have only shown this considering a subset of genes (i.e. expression matched group where there are 262 genes per group). The reviewer has compared these distributions to Figure 2A and erroneously inferred that there is a significant shift. In Figure 2A, we show the whole set of Active and PRCa genes (5,428 genes in total).

We apologize for not clearly highlighting this in the previous rebuttal letter. Now, in the revised version, we show distributions of noise and burst frequency levels across different expression cutoffs: from a cutoff of 10 mean normalized counts to 100. This is now directly comparable to Figure 2A. This analysis implies that there is highly significant difference between Active and PRCa genes in terms of noise and kinetic parameters, and that our results are robust to changes in selection of expression cutoff. Now, in the revised version, we have added distributions of DM and BF at different cutoffs as **Supplementary Figure 3A** as shown below:

Figure S3

A

Supplementary Figure 3. (A) Distribution of DM and burst frequency levels across different cutoffs of gene expression. Two-sided Wilcoxon rank-sum test P-values for differences of DM between Active and PRCa genes are:

$P < 2.2 \times 10^{-16}$, $P < 2.2 \times 10^{-16}$, $P < 2.2 \times 10^{-16}$, $P = 6.7 \times 10^{-15}$, $P = 2.3 \times 10^{-13}$, $P = 1.8 \times 10^{-13}$, $P = 8.1 \times 10^{-12}$, $P = 1.7 \times 10^{-13}$, $P = 2.2 \times 10^{-12}$ and $P = 5.6 \times 10^{-11}$ for gene expression cutoffs 10, 20, 30, 40, 50, 60, 70, 80, 90 and 100, respectively. For burst frequency levels, all P-values are $P < 2.2 \times 10^{-16}$.

Additionally, we have now added a plot showing how P-values change across different expression cutoffs (from a cutoff of 10 mean normalized counts to 100) for expression-matched groups of Active and PRCa genes. We should note that the increase in P-value is mainly due to the decreasing sample size leading to less statistical power. We have now added this as **Supplementary Figure 3C** as displayed below:

Figure S3

C

Supplementary Figure 3. (C) $-\log_{10}$ P-values are shown for differences between DM levels and BF levels of expression-matched Active and PRCa groups across different expression cutoffs. The

number of PRCa genes (that are expression-matched to same number of Active genes) are 666, 603, 540, 479, 427, 374, 341, 304, 280 and 262 for expression cutoffs of 10, 20, 30, 40, 50, 60, 70, 80, 90 and 100, respectively.

To reiterate, the choice of 10 reads/cell is an inappropriate, cutoff, since when taking the total reads per cell into account, this represents ~1 molecule per cell. This can hardly be considered to be a moderately to highly expressed gene. Even multiplying this by 10 does not reach a threshold that can be considered to be immune from technical noise in scRNA-seq experiments.

Using our 2i mESC scRNA-seq data with ERCC spike-ins (Kolodziejczyk et al. Cell Stem Cell. 2015), we find that the cutoff of 100 normalized counts per cell corresponds to 16.57 transcripts per cell (Please see **Rebuttal Figure 1** below). In our previous rebuttal, we have shown that the fraction of technical noise at the cutoff of 100 normalized counts saturates to 20% of technical noise on average. The 20% of technical noise can be considered as the minimum achievable technical noise level, and therefore the cutoff of 100 is a reasonable threshold minimizing the effect of technical noise. At this threshold, we have shown that our key result regarding Figure 2A is still statistically significant (P-value for DM: 5.6×10^{-11} , P-value for burst frequency: $< 2.2 \times 10^{-16}$). Increasing the expression cutoff (say 1000 normalised counts per cells suggested by the reviewer) does not decrease the technical noise level further and makes it impossible to compare the noise levels of genes between PRCa and Active since the sample size becomes too small.

Rebuttal Figure 1. Gene expression cutoff of 100 normalised counts per cell corresponds to 16.57 transcripts per cell according to our 2i mESC scRNA-seq data with ERCC spike-ins (Kolodziejczyk et al. Cell Stem Cell. 2015).

3. The response states that RNA FISH and single cell RNA-seq results consistently agree, and therefore there is no need for external validation. This is again an inaccurate statement. Numerous studies have described the extensive technical biases introduced by amplification of single cell RNA-seq data, which could explain the results the authors see here, but are not present in RNA FISH.

We disagree with this statement. In contrast, many studies have indicated that RNA FISH and single cell RNA-seq results correlate well. For example, in one of the pioneering single-cell studies, Shalek et al. (*Nature*, 2013) have shown that for 25 selected genes covering a wide range of expression levels, RNA-FISH closely mirrored the heterogeneity observed in the sequencing data (please see Fig.1d-g and Supplementary Fig2 in their paper). In Kumar et al. (*Nature*, 2014), gene expression distributions were also validated by smFISH and a good level of correlation for expression and coefficient of variation were observed for 14 selected genes in mESCs cultured in serum+lif (please see Fig. 1b and Extended Data Fig. 3 in their paper). Additionally, in another study, noise levels obtained by these two methods were found to be correlated in human fibroblast cells (Padovan-Merhar et al. *Molecular Cell*, 2015).

We agree that there would be amplification biases in single cell RNA-seq data for lowly expressed genes, and we analyse sensitivity and specificity in depth in our power analysis of scRNA-seq protocols available online and accepted in principle in *Nature Methods*:

<http://biorxiv.org/content/early/2016/09/08/073692>

In the work in the present manuscript, we are using a highly sensitive and specific SMART-seq protocol in Fluidigm C1 microfluidic platform, with a saturating sequencing depth of over a million reads per cell. This is one of the most sensitive and accurate scRNA-seq pipelines available, including microfluidic cell capture and full length transcript coverage. Moreover, we are comparing set of genes within this data set with comparable technical noise contribution, between PRC and non-PRC marked gene sets. In addition, our noise measure “DM” corrects for gene expression levels.

The authors cite Grun et al., to suggest that the methods agree. Grun et al of course shows

that single cell RNA-seq data is far less sensitive than RNA-FISH. However, more importantly, the data in Grun et al. contained unique molecular identifiers, which control for bias in PCR amplification, enabling them to match characteristics of the two distributions. The data here do not include UMIs, and so the comparison is both invalid and inaccurate.

We agree that RNA-FISH is more sensitive compared to single cell RNA-seq, and indeed Grun et al calculate the difference in capture efficiency for ERCC spike-ins versus endogenous mRNA using their CEL-seq protocol in well plates. We also show this in Supplementary Figure 2E in Svensson, Natarajan et al. *Nature Methods*, accepted (and available online in the BioRxiv link mentioned above). The CEL-seq protocol used by Grun et al. is different from the microfluidic cell capture plus SMART-seq protocol used in our work, where the sensitivity is approaching 1 mRNA per cell (Svensson, Natarajan et al. *Nature Methods*, accepted and available online in the BioRxiv link mentioned above).

The more important issue is actually accuracy of quantification of gene expression, which is imperfect for all RNA-seq methods, and is most likely better for smFISH (though there isn't a gold standard for smFISH calibration of accuracy of gene expression quantification to our knowledge). In Svensson, Natarajan et al (*Nature Methods*, accepted, 2017), we show that microfluidic SMART-seq has an accuracy of 0.88 in terms of the Pearson correlation coefficient between spike-in levels and read density (Figure 3A, <http://biorxiv.org/content/early/2016/09/08/073692>).

Additionally, we performed our noise analysis (DM) using single-cell RNA sequencing data of Grun et al. (serum and 2i ES cells). Comparison of global levels of gene expression heterogeneity in serum and 2i ES cells of Grun et al. (with UMIs) and Kolodziejczyk et al. (without UMIs) showed that differences in global gene heterogeneity profiles are not significant (Two-tailed paired t-test, please see Rebuttal Figure 2 below), indicating that the microfluidic SMART-seq protocol (without UMIs) is as efficient at deciphering cell-to-cell variation as CEL-seq with UMIs.

Serum ES cells

2i ES cells

Rebuttal Figure 2. Global gene heterogeneity profiles (quantified by our “DM” method) for Kolodziejczyk et al. and Grun et al. are similar for serum and 2i ES cells (Two-tailed paired t-test P-value=0.1 and P-value=0.2 for serum and 2i cells, respectively) indicating that the microfluidic SMART-seq protocol (without UMIs) is as efficient at deciphering cell-to-cell variation as CEL-seq with UMIs.

REVIEWERS' COMMENTS:

Reviewer #1 (Remarks to the Author):

The authors have addressed my previous comments,

Reviewer #2 (Remarks to the Author):

I think the authors have addressed all of the questions raised by Reviewer #3 now.

Reviewer #3 (Remarks to the Author):

The authors have addressed my primary concerns with the additional analyses in the revised manuscript. I appreciate the effort and care that the authors put into these revisions, and hope they agree that this has strengthened the manuscript.